# Pharmaceuticals as Emerging Pollutants in the Reclaimed Wastewater Used in Irrigation and Their Effects on Plants, Soils, and Groundwater

**Ghaida Ali Abdallat** [1] **, Elias Salameh** [1] **, Musa Shteiwi** [2] **and Sanaa Bardaweel** [3,*]

1 Department of Geology, University of Jordan, Amman P.O. Box 11942, Jordan;
ghidaaabdallat@yahoo.com (G.A.A.); salameli@ju.edu.jo (E.S.)
2 Department of Sociology, University of Jordan, Amman P.O. Box 11942, Jordan; m.shteiwi@ju.edu.jo
3 Department of Pharmaceutical Sciences, School of Pharmacy, University of Jordan,
Amman P.O. Box 11942, Jordan
* Correspondence: s.bardaweel@ju.edu.jo; Tel.: +962-775-190-831

**Abstract:** Pharmaceuticals and personal care products (PPCPs) were investigated in five wastewater treatment plants (WWTPs), groundwater, irrigated soils, and plants in Amman and Al-Balqa governorates in Jordan. PPCPs were extracted from water samples by solid-phase extraction (SPE) and analyzed by high-performance liquid chromatography coupled with tandem mass spectrometry (HPLC–MS/MS). Carbamazepine, ciprofloxacin, ceftiofur, diclofenac, erythromycin, lincomycin, ofloxacin, pyrimthamine, spiramycin, sulfamethoxazole, sulfapyridine, testosterone, trimethoprim, and thiamphenicol were detected in all raw wastewaters in μg/L, whereas 45 PPCPs were below the detection limits (<0.02 μg/L) in all samples. Na'ur and Abu Nuseir WWTPs showed high PPCPs removal efficiencies in comparison with AL-Baqa'a, Salt, and Fuhais-Mahis WWTPs. Boqorreya spring showed signs of contamination by Salt WWTP effluents as a result of mixing. Irrigation with effluents showed higher carbamazepine concentrations in soils at the top soil layers (0 to 20 cm) in all farms than its concentrations at the root zone (20 to 40 cm) by using drip irrigation system with various plants. In plants, carbamazepine concentration was only detected in high concentration level in mint leaves. In the same farm, diclofenac concentration was detected only in olives and not in twigs and leaves, indicating a high rate of plant uptake especially during the olive's growth period. Furthermore, plant fruits, leaves, and stems left on the farm after harvesting are generally consumed by cattle, which means entering the food chain of humans.

**Keywords:** pharmaceutical residues; wastewater treatment plants; effluents; reuse; irrigation

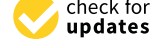



## 1. Introduction

Reuse of treated wastewater (TWW) in irrigation has become a widespread practice in countries in the Middle East and North Africa (MENA). Jordan, since the early 1980s, has worked to manage irrigation with treated wastewater (TWW) to be either discharged to mix with freshwater in dams and rivers and used for irrigation or to be directly used in irrigation without mixing. In Jordan, the main concern with the reuse of treated wastewater is its composition (chemicals, nutrients, and pathogens), which can cause environmental problems, such as eutrophication, groundwater deterioration, transmission of disease, as well as contamination of soils and plants. Therefore, Jordan issued and applied its first national standards and guidelines in 1995 (JS 893/1995) for different reuse applications of treated wastewater (irrigation, artificial groundwater recharge, and discharge to wadis or streams). In 2002, national standards were updated (JS 893/2002), prohibiting the use of treated wastewater for irrigation of vegetables eaten raw or recharging aquifers for potable use. In 2006, further revisions took place, providing less restriction for BOD$_5$, COD, and *E. coli* than what was stated in the previous guidelines (JS 893/2006) [1]. Table 1 shows the

Jordanian standards (JS 893/2006) for treated wastewater reuse in irrigation consisting of four categories (A, B, C, and D), which shows the suitable quality for the different crops. Moreover, the JS 893/2006 guidelines control the major pollutants ($BOD_5$, COD, TN, $NH_4$, $NO_3$, TP, $PO_4$, and TSS) that can be useful nutrients in agricultural irrigation and ensure the public and environmental health.

**Table 1.** The Jordanian standards for treated wastewater reuse in irrigation (JS 893/2006) [1].

| Parameter | A [1] | B [2] | C [3] | D [4] |
|---|---|---|---|---|
| $BOD_5$ (mg/L) | 30 | 200 | 300 | 15 |
| COD (mg/L) | 100 | 500 | 500 | 50 |
| DO (mg/L) | >2 | - | - | >2 |
| TSS (mg/L) | 50 | 150 | 150 | 15 |
| pH | 6–9 | 6–9 | 6–9 | 6–9 |
| Turbidity (NTU) | 10 | - | - | 5 |
| $NO_3$-N (mg/L) | 30 | 45 | 45 | 45 |
| TN (mg/L) | 45 | 70 | 70 | 70 |
| *E. coli* (MPN/100 mL) | 100 | 1000 | - | <1.1 |
| Intestinal Helminth eggs (egg/L) | < or =1 | < or =1 | < or =1 | <1 |
| Grease, oils, and fats (mg/L) | 8 | 8 | 8 | 8 |

[1] Cooked vegetables, parks, playgrounds, and sides of roads within city limits. [2] Fruit trees, sides of roads outside city limits, and landscape. [3] Field crops, industrial crops, and forest trees. [4] Cut lowers.

However, pollutants of emerging concern (PEC), such as pharmaceutical compounds, biocides, and some industrial chemicals, are not included in the JS reuse standards. Recent sporadic analyses show that they are present in the treated wastewater, which requires more investigations in order to elucidate their treatment and removal and effects on plant, soil, and groundwater and thus in the water cycle, food chain, and public health. Most important overall is the role of the public and society in understanding the dangers of such chemical residues on their health and the way that society cooperates in minimizing the amounts of such chemical input into the raw wastewater.

Table 2 presents the common pharmaceuticals detected in wastewater [2–4]. The main sources of these emerging contaminants are pharmaceuticals and personal care products (PPCPs) from hospitals, households, industries, wastewater treatment plants, endocrine-disrupting chemicals (EDCs), plasticizers (e.g., bisphenol-A), flame-retardants, fuel additives, and other industrial organic products [5]. Worldwide, PPCPs have been detected in all environmental compartments, such as water, soil, air, biota, and wastewater, at concentrations ranging from sub-ng/L levels to g/L [6–8].

**Table 2.** Some of the common pharmaceuticals found in wastewater [3,4].

| Category | Pharmaceutical Compound | Category | Pharmaceutical Compound |
|---|---|---|---|
| Diagnostic Aid-adsorbable Organic Halogen Compounds | Iopromide Iomeprol | Antibiotics | Sulfamethoxazole, ciprofloxacin, Ofloxacin, Trimethoprim, Penicillin |
| Analgesic Antipyretics | Acetaminophen, Phenylbutazone, Naproxen, Ibuprofen, Ketoprofen | Anticancer | Cyclophosphamide, Ifosfamide, Daunorubicin, Tamoxifen, Letrozole, Methotrexate |
| CNS Drugs | Caffeine, Carbamazepine | Endocrinology Treatments | 17 α-ethinylestradiol, Estrone, 17 β-estradiol, Estriol |
| Cardiovascular Drugs | Propranolol, Atenolol, Betaxolol, Metoprolol, Clofibric Acid, Gemfibrozil, Fenofibrate | | |

Moreover, many studies indicate that the current wastewater treatment processes (primary and secondary) cannot eliminate all PPCPs found in the raw wastewater [9,10]. Therefore, municipal wastewater treatment plants (WWTPs) are considered as a main source for the discharge of PPCPs into surface waters [11]. Many studies have shown that anthropogenic micro-pollutants, including pharmaceuticals (e.g., carbamazepine, diclofenac, gabapentin), artificial sweeteners (e.g., acesulfame), X-ray contrast media (e.g., iohexol, Iopromide), or corrosion inhibitors (e.g., benzotriazole), are only incompletely removed in conventional wastewater treatment processes and are subsequently widespread in the aquatic environment [7,12–14].

The purpose of our study is to assess the existing PPCPs in Na'ur, Fuhais-Mahis, Abu-Nuseir, and Al Baqa'a wastewater treatment plants as major effluent discharge sites and track the mixing process along wadis with springs' water until reuse application in irrigation to assess the risk potential to the environment. Furthermore, this study aimed to improve the scientific understanding of the biodegradability of PEC and the other agro- and industrial chemicals originating from TWW in soils and during movement through the unsaturated zone to the groundwater and the uptake of PEC originating from TWW by edible plants in the semi-arid region in addition to elaborating recommendations for the safe reuse of TWW in agriculture. Pharmaceuticals have not yet been considered within the Jordanian standards for reuse because they are considered as recently emerging pollutants. It is hoped that this study will provide awareness for policymakers to improve the governance and treatment of such substances before reaching wastewater treatment facilities.

## 2. Medicine Consumption in Jordan and Its Relationship with Social Life Practices

Pharmaceutical manufacturing is a well-established sector in Jordan that covers about one-quarter of the country's needs of medicines [15]. All types of medicines should be authorized and registered by the Jordan Food and Drug Administration (JFDA) before being released to the market. The JFDA is a regulatory and an independent institution that approves the safety, quality, and affordability of medicines [16].

Both the social patterns of behavior and the belief system related to health and sickness, and therefore patterns of consumption of medication and their discharge, become important in understanding the presence of pharmaceuticals in wastewater treatment. First, it is a common practice that there is an overuse of antibiotics among the population due to the patterns of seeking medical help [17]. In Jordan, people go to pharmacies to seek medical help instead of seeing physicians, especially for seasonal colds and infections. The ease of antibiotic prescriptions and the lack of strict supervision of pharmacy prescriptions most likely would result in over-prescription of antibiotics and other medications [17]. Secondly, the lack of medical awareness among people may lead to self-termination of medications once feeling well or switching between medicines without obtaining professional advice [18]. Therefore, it is likely that there will be an unjustified discharge of unused medicine. Thirdly, it safe to assume that there is a lack of awareness regarding the proper and safe method of discharging the medications among most of the population. These above social and cultural factors are important in understanding the problem with clear policy implications.

The most common diseases occurring among the Jordanian people are chronic diseases, such as hypertension, diabetes mellitus, inflammatory diseases, and respiratory diseases [19]. About one-third of the Jordanian population over the age of 25 years have at least one of the mentioned chronic diseases. Accordingly, it is expected to see high consumption of chronic diseases medications among the Jordanian people.

In addition, irrational and unjustified prescribing patterns and polypharmacy occurrence among the Jordanian people have been previously reported [18]. For example, in Jordan, self-medication with antibiotics is very common within the population [17]. Usually, patients acquire antibiotics from community pharmacies and medication leftovers from prior prescriptions. Such practice of self-medication has resulted in the appearance of antibiotic resistance within the Jordanian community.

## 3. Materials and Methods

### 3.1. Study Area

The study area covers five wastewater treatment plants and spring and reuse sites irrigated by treatment plants' effluents in Amman and Al-Balqa governorates (high population density). Figure 1 shows the wastewater treatment plants with the nearby spring and reuse sites where pharmaceuticals and other pollutants were investigated. The first facility is Na'ur wastewater WWTP, which receives wastewater from Na'ur and Wadi Sir cities and from Al Hussein Medical Center (the biggest medical center in the country). The other sites of Fuhais-Mahis, Salt (Shueib), Abu-Nuseir, and Baqa'a WWTPs are where the reuse in irrigation is applied. Table 3 summarizes the characteristics of these plants, the treatment technology methods, design flow capacity, inflow and outflow quantities, and reuse applications.

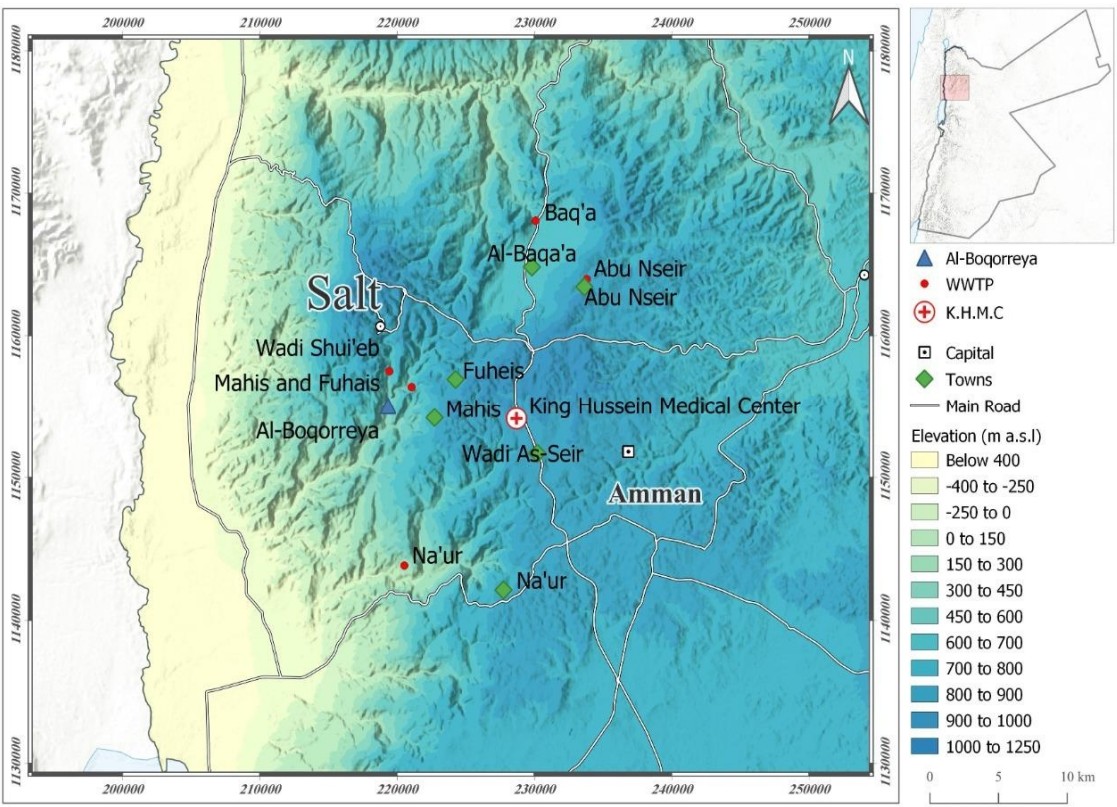

**Figure 1.** Location of the study area showing the wastewater treatment plants and Boqorreya spring sites. Sites of soil and plant samples lie in the downstream area of WWTPs.

### 3.2. Sampling, Sample Preparation, and Extraction

Ten influent and effluent wastewater samples, two water mixing points (effluents with fresh springs), ten soil samples, and eleven planted fruit samples were collected during summer after the COVID-19 pandemic during six sampling events (26th and 30th August and 1st, 3rd, 8th, and 20th September 2020) for the analyses and investigation of the physico-chemical properties (ionic speciation, charge, functional groups) of PEC and the other agro- and industrial chemicals that are most influential in term of uptake and translocation.

**Table 3.** Characteristics of the studied wastewater treatment plants and their operation conditions and reuse applications [20].

| WWTP | Treatment Method | Capacity | Influent | Effluent | Reuse Applications (MCM/Year) | |
| | | (MCM */Year) | | | Irrigation | Discharge to Wadis |
|---|---|---|---|---|---|---|
| Baqa'a | Trickling filter | 5.4 | 5.3 | 5.1 | 0.4 | 4.7 |
| Salt (Shueib) | Activated sludge | 2.8 | 3 | 2.8 | 1 | 2.8 |
| Fuhais-Mahis | Activated sludge | 1.5 | 1.17 | 1 | 0.3 | 0.7 |
| Abu-Nuseir | Activated sludge | 1.5 | 1.2 | 1.2 | 0.2 | 1 |
| Na'ur | Mechanical, chemical, and biological | 17.5 | 1.8 | 1.8 | 0.8 | 1 |

* MCM, million cubic meters.

Direct field measurements (electrical conductivity (EC), pH, and temperatures (T °C)) were conducted onsite for all water samples. The samples were analyzed at the Royal Scientific Society (RSS) (pharmaceuticals lab) using high-performance liquid chromatography combined with tandem mass spectrometry (HPLC/MS/MS). The water samples were collected in 1 L plastic bottles rinsed with methanol and then washed with type I purified reagent water. All samples were stored in a refrigerator under dark conditions at 4 to 8 °C. The extraction process was implemented according to the procedure provided by Water Sciences Laboratory at the University of Nebraska–Lincoln (WSL/UNL) in the United States (USA) [21]. Then, samples were pre-concentrated using solid-phase extraction (SPE) after sampling (within 24 h). The first step was removing suspended particles and then filtering using 0.45-micron glass-fiber filters with vacuum filtration unit. The filtered sample was then pumped via tube to the cartridge using a vacuum manifold system. The sample flow through the SPE cartridge was kept at approximately 10 mL/min or less. After the whole sample was extracted, the cartridge was rinsed with 5 mL of DDI $H_2O$. Room air was allowed to flow through the cartridge by continued suction for a minimum of 5 min to help dry the cartridge. Sample cartridges were analyzed using liquid chromatography tandem mass spectrometry (LC-MS/MS), following the standards.

Soil samples were collected at different depths at 20 (top soil layer) and 40 (root zone) cm in plastic bags (500 mg each sample). Soil pastes were prepared and filtered in order to ensure the soil extraction process was implemented according to the procedure provided by Water Sciences Laboratory at the University of Nebraska–Lincoln (WSL/UNL) in the United States (USA) [21].

Plant samples were separated into roots, leaves, and fruits during collection. In case of mint, the samples were not separated during collection. Before extraction of the plant material, their parts were rinsed with tap water, freeze-dried, and ground to fine powder. Extraction of target compounds from plant samples was conducted using solid–liquid extraction (SLE).

## 4. Results

### 4.1. Field Measurements (EC, pH, and T (°C))

The treatment performance of the wastewater treatment plants was evaluated by the conformity of effluents to the national JS (class A) for reuse in irrigation. Table 4 summarizes the results of field measurements (EC, pH, and T (°C)) over the study period. The effluents pH values ranged from 6.45 to 7.83, which is conformed to the JS (pH values of 6–9). There was no significant change in pH values during the treatment process compared with raw wastewater pH, indicating a normal biological and chemical process in the wastewater treatment plants. On the other hand, Boqorreya spring showed the highest pH value (8.0), and the pH values were reduced to 7.85 directly at the mixing point with Salt effluents.

**Table 4.** Field measurements of the WWTPs and other sampling points along wadis in the study area.

| Name of Water Sample | EC (µs/cm) | pH | T (°C) |
|:---|:---:|:---:|:---:|
| Salt WWTP (Raw) | 1871 | 7.47 | 28.2 |
| Salt WWTP (Effluent) | 1693 | 7.34 | 29.1 |
| Boqorreya Spring | 617 | 8.01 | 30.8 |
| Mixed water (Salt outlet + Boqorreya Spring) | 1039 | 7.85 | 32.0 |
| Fuhais-Mahis WWTP (Raw) | 1790 | 7.71 | 27.2 |
| Fuhais-Mahis WWTP (Effluent) | 1459 | 7.80 | 27.5 |
| Na'ur WWTP (Raw) | 1765 | 7.63 | 30.0 |
| Na'ur WWTP (Effluent) | 1434 | 7.31 | 30.0 |
| Al-Baqa'a WWTP (Raw) | 2390 | 7.35 | 27.8 |
| Al-Baqa'a WWTP (Effluent) | 1859 | 7.83 | 25.2 |
| Abu-Nuseir (Raw) | 1945 | 8.00 | 25.6 |
| Abu-Nuseir (Effluent) | 1396 | 6.45 | 25.7 |

The effluent EC was reduced gradually during the treatment process in all the WWTPs. The EC results conform to the JS (0.7–3000 µS/cm) over the study period. The results of the analyses show that the tested wastewater treatment plants produce treated wastewater that conforms with the Jordanian standards (JS) for reuse in irrigation for all the common parameters, which are, until now, used worldwide as indicators of effluent quality for reuse in irrigation and where the new and upcoming pollutants were not considered.

*4.2. PPCPs in WWTPs*

Table 5 lists the detected 14 PPCPs and their variations in the collected samples in µg/L. These compounds are carbamazepine, ciprofloxacin, ceftiofur, diclofenac, erythromycin, lincomycin, ofloxacin, pyrimthamine, spiramycin, sulfamethoxazole, sulfapyridine, testosterone, trimethoprim, and thiamphenicol. However, 45 PPCPs were below the detection limits (<0.02 µg/L) in all samples.

Figures 2–4 show the concentrations of higher PPCPs that measured in the WWTPs influent and effluent water samples, Boqorreya spring water, and at the mixing point of Salt WWTP effluents with Boqorreya Spring water in µg/L. Carbamazepine was the highest concentration of pharmaceutical residual that was found in all water samples (influents (raw), effluents, and spring water) as shown in Figure 2a. However, low carbamazepine concertation was measured at 0.03 µg/L in Boqorreya spring as a fresh groundwater source, indicating little signs of contamination via percolation of carbamazepine from the mixing point. That carbamazepine concentration was measured of 0.43 µg/L at the mixing point of Boqorreya spring with Salt effluents.

Moreover, diclofenac as a common anti-inflammatory medicine (its brand name Voltaren) was detected and measured in all raw wastewater, partly in effluent water samples and in the mixing point of Salt and Boqorreya spring waters (Figure 2b). The concentrations of diclofenac were eliminated through treatment process at Abu-Nuseir and Na'ur WWTPs. However, it concentrations were partly removed by Salt, Fuhais-Mahis, and Al-Baqa'a WWTPs as a result of low biodegradation during biological wastewater treatment, whereas diclofenac concentration in Al-Baqa'a WWTP effluent was 2.6 µg/L higher than the recommended level of less than 1 µg/L as reported by Vieno and Sillanpää [22].

**Table 5.** Concentration of PPCPs in various WWTP water samples, Boqorreya spring, and mixing point of Salt WWTP effluents with Boqorreya Spring water in (µg/L).

| | Salt WWTP (Raw) | Salt WWTP (Effluent) | Boqorreya Spring | Mixed Water (Salt Effluent + Boqorreya Spring) | Fuhais-Mahis WWTP (Raw) | Fuhais-Mahis WWTP (Effluent) | Na'ur WWTP (Raw) | Na'ur WWTP (Effluent) | Al-Baqa'a WWTP (Raw) | Al-Baqa'a WWTP (Effluent) | Abu-Nuseir (Raw) | Abu-Nuseir (Effluent) |
|---|---|---|---|---|---|---|---|---|---|---|---|---|
| Carbamazepine | 0.97 | 1.13 | 0.03 | 0.43 | 4.45 | 1.37 | 1.38 | 0.7 | 1.31 | 0.71 | 2.11 | 0.55 |
| Ceftiofur | <0.02 | <0.02 | <0.02 | <0.02 | <0.02 | <0.02 | <0.02 | <0.02 | 0.03 | <0.02 | <0.02 | <0.02 |
| Ciprofloxacin | 2.39 | <0.02 | <0.02 | <0.02 | 1.0 | 0.02 | 1.43 | <0.02 | 1.4 | 0.15 | 1.9 | <0.02 |
| Diclofenac | 1.69 | 0.87 | <0.02 | 0.28 | 1.39 | 0.84 | 1.86 | <0.02 | 2.44 | 2.64 | 2.88 | <0.02 |
| Erythromycin | 0.03 | 0.05 | <0.02 | 0.02 | <0.02 | <0.02 | <0.02 | <0.02 | 0.04 | <0.02 | 0.1 | 0.02 |
| Lincomycin | 3.07 | 0.02 | <0.02 | 0.02 | 1.04 | 0.04 | 2.29 | <0.02 | 3.32 | 0.86 | 1.22 | <0.02 |
| Ofloxacin | 0.22 | 0.05 | <0.02 | 0.02 | 0.44 | 0.13 | 0.48 | <0.02 | 0.49 | 0.1 | 0.67 | <0.02 |
| Pyrimethamine | 0.66 | 0.22 | <0.02 | 0.09 | 0.9 | 0.08 | 0.34 | <0.02 | 0.68 | 0.07 | 0.77 | <0.02 |
| Spiramycin | 1.07 | 0.44 | <0.02 | 0.09 | 0.73 | 0.1 | 2.66 | <0.02 | 0.84 | 0.52 | 1.26 | <0.02 |
| Sulfamethoxazole | 2.08 | 0.43 | <0.02 | 0.12 | 0.36 | 0.09 | 0.72 | 0.02 | 1.38 | 0.27 | 0.02 | 0.02 |
| Sulfapyridine | 0.72 | 0.23 | <0.02 | 0.09 | 0.91 | 0.08 | 0.34 | <0.02 | 0.68 | 0.06 | 0.71 | <0.02 |
| Testosterone | 0.05 | <0.02 | <0.02 | <0.02 | 0.08 | <0.02 | 0.18 | <0.02 | 0.05 | <0.02 | 0.09 | <0.02 |
| Trimethoprim | 0.13 | <0.02 | <0.02 | <0.02 | <0.02 | <0.02 | 0.03 | <0.02 | 0.11 | 0.03 | <0.02 | <0.02 |
| Thiamphenicol | <0.02 | <0.02 | <0.02 | <0.02 | <0.02 | <0.02 | <0.02 | <0.02 | <0.02 | <0.02 | <0.02 | <0.02 |

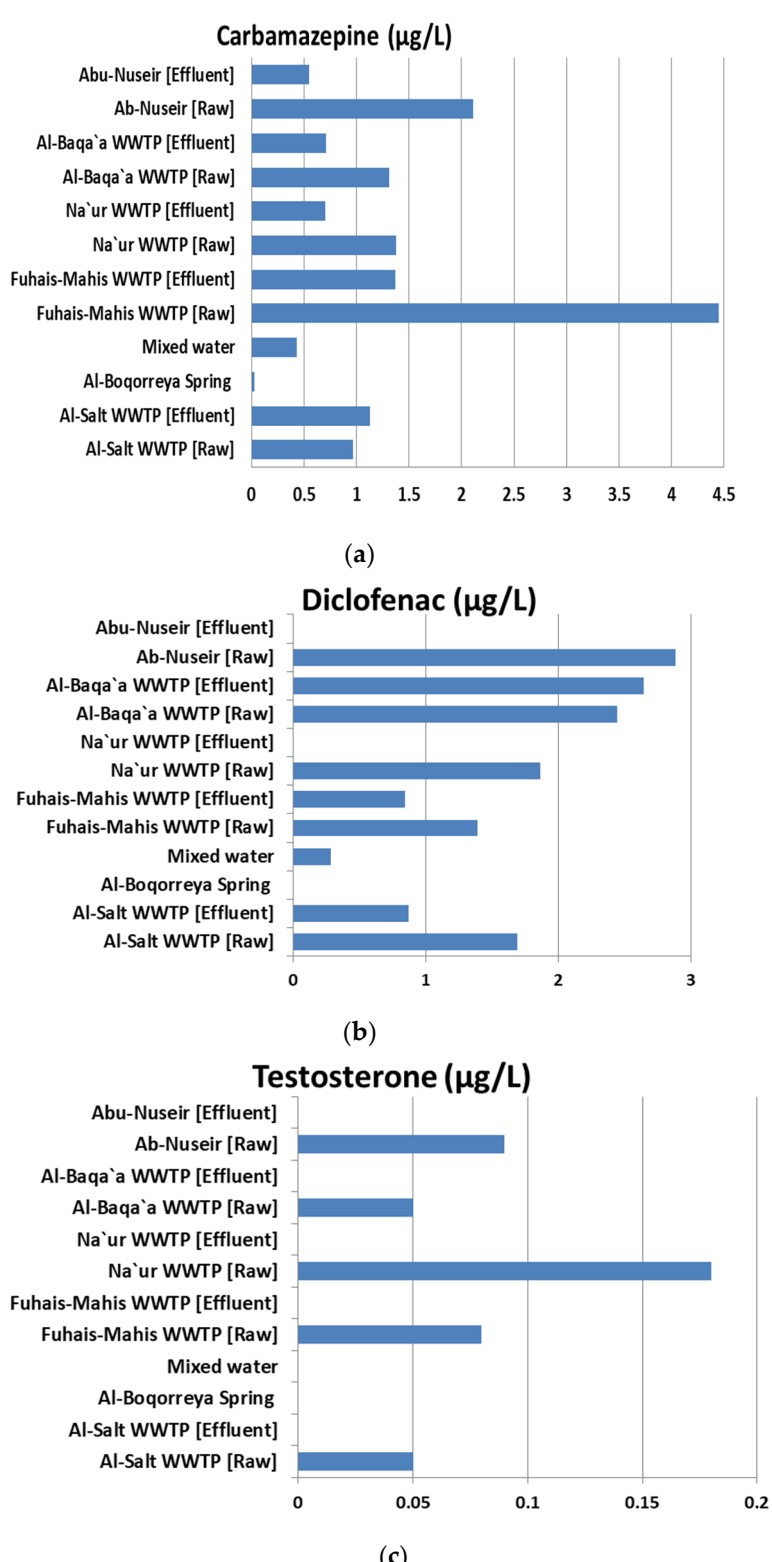

**Figure 2.** Concentration of: (**a**) Carbamazepine concentrations, (**b**) Diclofenac concentrations, and (**c**) testosterone hormone concentrations of the investigated water samples.

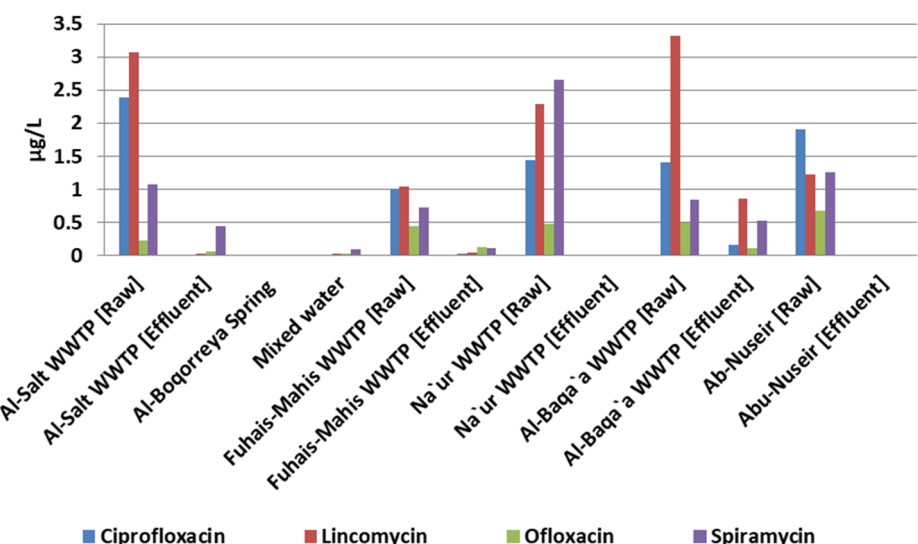

**Figure 3.** Concentration of lincomycin, ciprofloxacin, ofloxacin, and spiramycin in the investigated water samples.

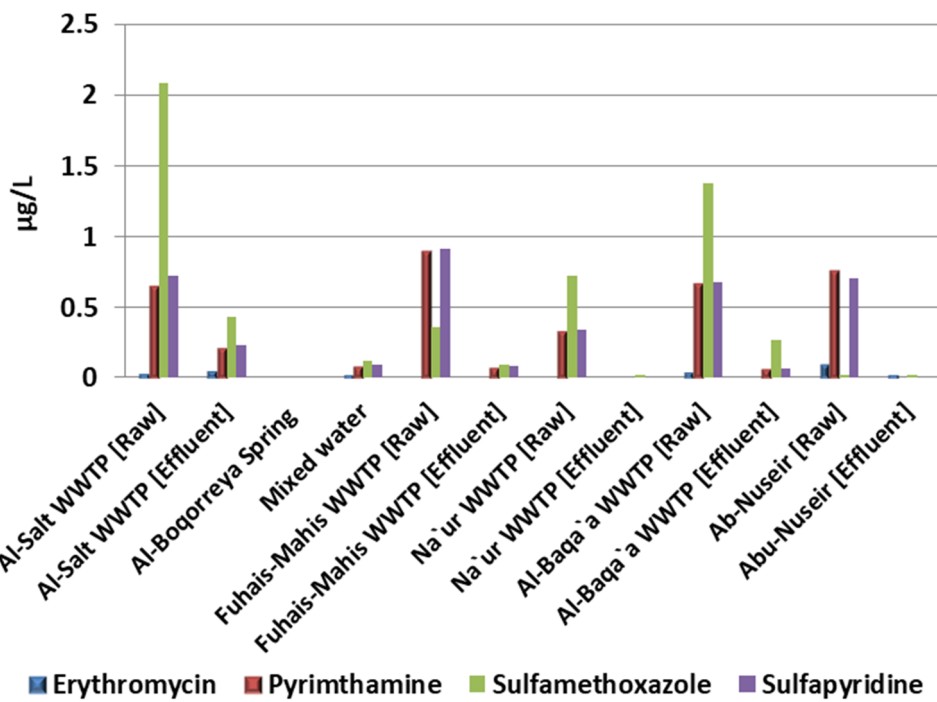

**Figure 4.** Concentration of sulfapyridine, sulfamethoxazole, pyrimthamine, sulfapyridine, and erythromycin in the investigated water samples.

Furthermore, testosterone hormone was detected only in the raw wastewater samples, and the results indicated that testosterone is eliminated by WWTPs (Figure 2c). Therefore, there is no potential of adverse environmental and public health impact.

Seven antibiotics were detected in the raw and treated wastewater of 59 PCPs in the WWTPs. These were lincomycin, ciprofloxacin, ofloxacin, ceftiofur, erythromycin, spiramycin, sulfamethoxazole, sulfapyridine, trimethoprim, and thiamphenicol (Figure 3). Lincomycin as an antibiotic was the second-highest detected compound, especially in all raw wastewater of the WWTPs, whereas its concentrations were reduced in Al-Baqa'a, Salt, and Fuhais-Mahis effluents and totally removed through treatment process in Na'ur and

Abu-Nuseir WWTPs (Figure 3). Generally, antibiotics partition into water is based on the chemical and physical properties of the antibiotic itself [23].

Ciprofloxacin and ofloxacin (fluoroquinolones class) showed notable level concentrations in all wastewater influents (Figure 3). However, ciprofloxacin concentrations were not detected in Salt, Na'ur, and Abu-Nuseir effluents, presenting high removal efficiency through their treatment process and were highly reduced by Fuhais-Mahis and Al-Baqa'a WWTPs effluents. This can be explained by the various retention time that each WWTP provided during treatment as reported by [24]. However, ofloxacin concentrations were reduced in Salt WWTPs, and therefore, it was detected in the mixing point of Boqorreya and treated wastewater.

Spiramycin was highly eliminated by Na'ur, Abu-Nuseir, and Fuhais-Mahis WWTPs. Similar results were reported by Lofrano and others [25] that this compound showed high removal by activated sludge treatment plants more than 99.9% during summer and about 9% during winter, whereas Salt WWTP showed less removal efficiency, among others, which affected the mixing water point of Salt effluents and Boqorreya spring.

Ceftiofur antibiotic was detected only in Al-Baqa'a treatment plant influents with 0.03 μg/L, whereas it was measured 0.13 and 0.03 μg/L in Salt and Na'ur WWTPs effluents, respectively. Additionally, it was highly removed through its treatment process.

Sulfamethoxazole, sulfapyridine, and erythromycin antibiotics were detected in influents and some effluents of WWTPs and in the mixing point of Al-Salt effluents with Boqorreya fresh water as depicted in Figure 4.

### 4.3. PPCPs in Cultivated Farms

PPCPs were investigated in three accessible farms where reuse in irrigation is highly applied in Salt and Fuhais-Mahis areas using drip irrigation systems. In irrigated soils, carbamazepine was the only detected compound among 58 measured PPCPs. Figures 5 and 6 present CBZ concentrations in irrigated soils where higher carbamazepine concentrations were measured at the top soil layers (0 to 20 cm) in all farms in comparison with lower carbamazepine concentrations at the root zone (20 to 40 cm) with various plants. Plant uptake is negligible in farms A and B due to no PPCPs measurements in plants and fruit. Thus, soil organic matter is considered as a sorbent for carbamazepine, reducing its concentration in the soils and therefore hindering its bioavailability for plant uptake [26]. On the other hand, carbamazepine concentrations in soils irrigated with treated water were higher than its concentration in treated water as a result of accumulation over long-term reuse. However, the farm that irrigated from Fuhais-Mahis effluent presents higher carbamazepine accumulation in soil layers, indicating continuous accumulation in soils that can be varied by soil physical and chemical properties. Borgman and Chefetz [27] reported that transport of PPCPs can be hindered by high loading of organic matter in soil columns.

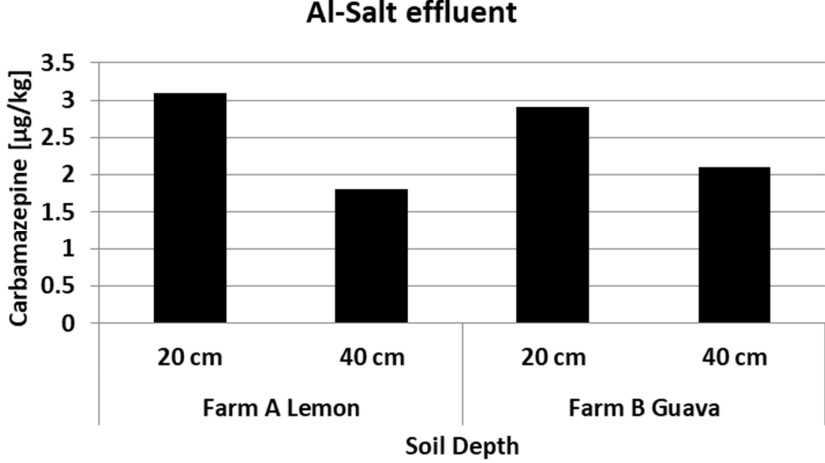

**Figure 5.** The concentration of carbamazepine in two different cultivated farms in Wadi Shueib.

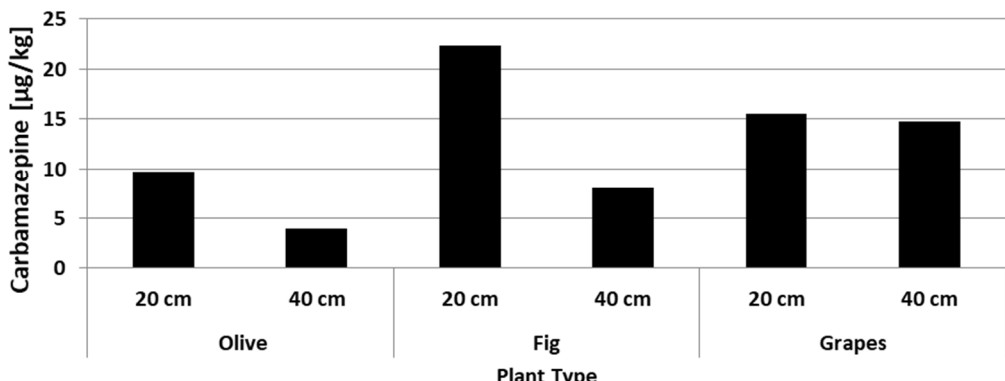

**Figure 6.** The concentration of carbamazepine in soils cultivated with various plants and irrigated by Fuhais-Mahis effluents in Mahis city.

Two of 58 PPCPs were detected in plants parts. Figure 7 shows the concentration of carbamazepine in mint plants and diclofenac concentration in olive. According to the JS for reuse of treated wastewater in irrigation, it is prohibited to irrigate leaves of plants grown in soils, such as mint, which is found on farm C. Similar to findings reported by Goldstein et al. [28] and Wu et al. [29], the carbamazepine concentration was much higher in leaves than in fruit. In the same farm, concentration of diclofenac was detected in olives, whereas it was not detected in twigs and leaves, indicating a high rate of plant uptake, especially during the olive's growth period.

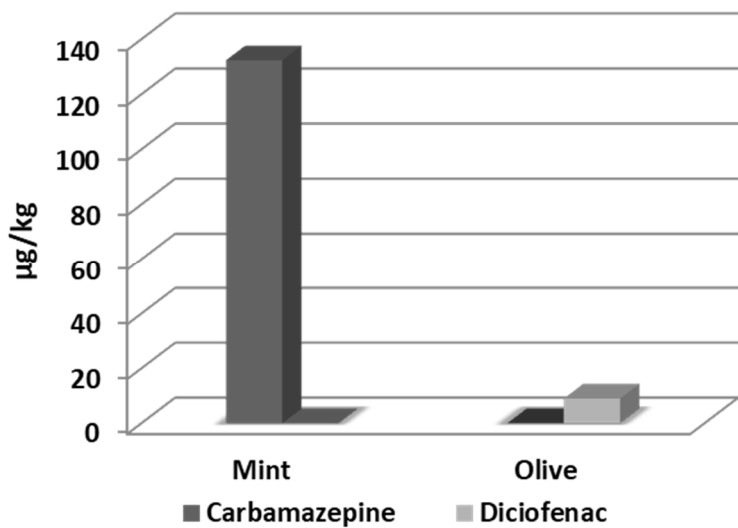

**Figure 7.** The concentration of carbamazepine and diclofenac uptake in mint leaves and olives irrigated with Fuhais-Mahis effluents.

## 5. Discussion

Most pharmaceutical residues are flushed into the sewer system via toilets, bathing water, and wastewater from washing machines [30], which illustrates higher concentrations of PPCPs in all WWTPs influents. However, the results showed that through high removal efficiency of PPCPs, most of the detected compounds were reduced and eliminated through treatment process at the WWTPs. Similar results were documented by Al-Mashaqbeh et al. [11], indicating that the majority of pharmaceutical compounds in Jordan can be partially removed by the activated sludge system used at As-Samra WWTP. Furthermore, several studies have documented that the removal efficiency for pharmaceutical compounds can vary based on different applied wastewater treatment process [31].

The highest concentration of pharmaceutical residual in the study was carbamazepine, which was found in all water samples (influents (raw), effluents, and spring water). Carbamazepine, as an anticonvulsant and mood-stabilizing drug, is poorly removed by the conventional WWTPs (<10%) [32]. Similar results were reported by Al-Mashaqbeh et al. [11] that carbamazepine concentration was high in As-Samra WWTP influent (1100 ng/L) and effluent (850 ng/L), resulting in low removal of the substance (23%). In addition, many studies have demonstrated the correlation between using reclaimed water and its presence in groundwater [33]. On the other hand, carbamazepine concentration in Al-Baqa'a WWTP effluent has 2.6 µg/L higher than the recommended level of less than 1 µg/L as reported by Vieno and Sillanpää, [22]. Diclofenac is known to have harmful effects on several environmental species at concentrations of ≤1 µg/L [22]. Baqa'a WWTP showed high concentrations of ceftiofur antibiotic, which is used in cattle treatment and might be related to more cattle and markets for the trading of livestock in Al-Baqa'a area. Additionally, trimethoprim antibiotic was measured in Al-Baqa'a treatment plant influents (0.11 µg/L) and effluents (0.03 µg/L).

Various studies have found that antibiotics (specifically sulfamethoxazole) decompose during transport within water systems [34]. Additionally, sulfamethoxazole and sulfapyridine concentrations in the effluents were found probably due to the biotransformation of their acetylated forms in anaerobic treatment [35].

Plant uptake is negligible in the investigated farms due to the lack of PPCPs measurements in plants and fruit. Thus, soil organic matter is considered as a sorbent for carbamazepine, reducing its concentration in the soils and, as a consequence, hindering its bioavailability for plant uptake [26].

## 6. Conclusions and Recommendations

Treated wastewater, from the standpoint of concentrations of pharmaceutical, biocide, and industrial products, contains new pollutants in concentrations considered hazardous to health when used in irrigation because they enter into the plants' roots, stems, leaves, and fruits. This study explored the removal and transactions of PPCPs through WWTPs mixing with fresh water, irrigated soils, and plant uptake. To conclude, 14 PPCPs were detected and measured in the collected samples in µg/L. These compounds are carbamazepine, ciprofloxacin, ceftiofur, diclofenac, erythromycin, lincomycin, ofloxacin, pyrimthamine, spiramycin, sulfamethoxazole, sulfapyridine, testosterone, trimethoprim, and thiamphenicol. However, 45 PPCPs were below the detection limits (<0.02 µg/L) in all samples. Na'ur and Abu-Nuseir WWTPs showed high PPCPs removal efficiency in comparison with AL-Baqa'a, Salt, and Fuhais-Mahis WWTPs. Boqorreya spring showed small signs of contamination by Salt WWTP effluents. Because the treated effluents infiltrate to join the groundwater, depending on the mixing ratio with the groundwater, these pharmaceutical residues may represent a health detriment to humans and animals drinking that water.

The managerial conclusion is that Jordan, by now, needs new laws, by-laws, and regulations to govern the production, distribution, use, collection, and proper disposal of unused pharmaceutical residues along the whole chain of producers, distributers, users, and handers of the unused quantities. Three distinct policy recommendation are relevant here: First, there is a need for clear rules and regulations about discharging medical waste by all medical establishments that should also include monitoring of this process. Second, there is a need to have strict rules to limit the ability of pharmacies to prescribe certain medications without physician prescriptions, such as antibiotics. Finally, the proper ways of discharging unused medicine or expired medicine should be standardized. This is a governance issue, which involves the government, the parliament, and especially the society, and concerns awareness programs and social behavior of each individual dealing with pharmaceuticals.

**Author Contributions:** Conceptualization, G.A.A., E.S., S.B. and M.S.; formal analysis, G.A.A. and S.B.; funding acquisition, E.S. and M.S.; investigation, G.A.A.; methodology, G.A.A. and E.S.; project administration, M.S. and S.B.; resources, G.A.A.; software, G.A.A.; supervision, G.A.A., S.B., E.S. and M.S.; writing—original draft, G.A.A.; writing—review and editing, G.A.A., E.S., S.B. and M.S. All authors have read and agreed to the published version of the manuscript.

**Funding:** This research was funded by the Deanship of Scientific Research at the University of Jordan.

**Institutional Review Board Statement:** Not applicable.

**Informed Consent Statement:** Not applicable.

**Data Availability Statement:** The data that support the findings of this study are available from the corresponding author, [S.B.], upon reasonable request.

**Acknowledgments:** The authors express their appreciation to the Jordanian Ministry of Water and Irrigation for providing water-related information during the course of this work.

**Conflicts of Interest:** The authors declare no conflict of interest.

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
