# Peer review of "Pharmaceuticals as Emerging Pollutants in the Reclaimed Wastewater Used in Irrigation and Their Effects on Plants, Soils, and Groundwater"

_water, doi:10.3390/w14101560_

Round 1

Reviewer 1 Report

Comments on this paper are as follows.

1. Title and content do not match.
What are "Other emerging pollutants"?
2. "Plants, soils, and Groundwater, Human, and Animal Health" doesn't seem to have any results.
I've seen the results of the soil, but other results are just speculation.

3. Abbreviations for pharmaceutical substances are only partially used. If you use abbreviations, apply them all.

4. "Risk" was not assessed in this paper in line 80-88. Rewrite the "object" in this study.

5. In the title and purpose of the study, be sure to mention only the part that presents the results.

6. Add references to the sentences on lines 100-115.

7. 3.1. study area
Except for the location of the WWTP, the characteristics of the target point are unknown. Please add the characteristics of the target point.

8. In Table 3, what is the "MCM"?

9. 3.2. Sampling, sample preparation, and extraction
Please specify the sampling method, SPE method, and LC-MS/MS method. This is a very important part.

10. Redraw Fig 2-4 in the same format.
Why is the graph divided into 3 parts and why are the substances divided?

11. 4.3. PPCPs in cultivated farms
Add to "3.2", specifying "how to sample soil"

12. Explain why CBZ concentrations are high in line 298.

13. Why is only the CBZ concentration presented in Fig 6?

14. Why was the surface layer of soil 20 cm and 40 cm presented?

15. Why are all pharmaceuticals not presented in Fig 5?

16. The "discussion" session should fully explain the parts of the "results" session. Please explain in detail the high and low concentration of pharmaceutical substances and the source of surrounding pollution.

17. Why did Na'ur WWTP show high PPCPs removal efficiency in Table 5? explain it please.

18. There are many typos throughout the paper. Please correct it.

Author Response

Dear respected Editor of Water,

Thank you for your letter and constructive comments concerning our manuscript entitled Pharmaceuticals as Emerging Pollutants in the Re-Claimed Wastewater used in Irrigation and their Effects on Plants, Soils, and Groundwater . We have studied your comments carefully and made corrections which we hope would meet with your approval. Please consider our point-by-point responses in the following texts.

Response to Reviewer 1:

First, authors thank you for your comments, notes and suggestions.

Title and content do not match. What are "Other emerging pollutants"?

Our response:

The title of this study has been modified to “Pharmaceuticals as Emerging Pollutants in the Re-Claimed Wastewater used in Irrigation and their Effects on Plants, Soils, and Groundwater” in order match the study content.

Plants, soils, and Groundwater, Human, and Animal Health" doesn't seem to have any results.
I've seen the results of the soil, but other results are just speculation.

Our response:

We thank the respected reviewer for the point. Page 6, line 196, heading 4.2. PPCPs in WWTPs shows the concentrations of Pharmaceutical residues  in wastewater treatment plants, Boqorreya Spring (groundwater) and the mixed water [Salt outlet + Boqorreya Spring]. In addition, page 9,  line 261 shows 4.3. Results of PPCPs in Cultivated Farms, including soils and plants analyses (Figure 5, 6 and 7).

Abbreviations for pharmaceutical substances are only partially used. If you use abbreviations, apply them all.

Our response:

Abbreviations have been used in whole text in the revised version of the manuscript

"Risk" was not assessed in this paper in line 80-88. Rewrite the "object" in this study.

Our response:

We thank the respected reviewer for the valuable suggestion.

Line 80 has been rephrased into the following:  The purpose of our study is to assess the existing PPCPs in Na`ur, Fuhais-Mahis, Abu- Nuseir, and Al Baqa`a wastewater treatment plants as major effluent discharge sites and tracking the mixing process along wadis with springs` water until reuse application in irrigation to assess the risk potential to the environment. Furthermore, soils, plants and surface and groundwater are part of the environment. So the presence of PPCPs has been assessed through this study.

In the title and purpose of the study, be sure to mention only the part that presents the results.

Our response:

We thank the respected reviewer for this valuable comment. The title of this study has been modified into “Pharmaceuticals as Emerging Pollutants in the Re-Claimed Wastewater used in Irrigation and their Effects on Plants, Soils, and Groundwater”. Thus, surface, groundwater, plants and soils analyses are mentioned in the title and results.

Add references to the sentences on lines 100-115.

Our response:

References (17 and 18) were added in the modified version of the manuscript.

  1. Al-Azayzih, Ahmad, Rawan Alamoori, and Shoroq M. Altawalbeh. "Potentially inappropriate medications prescribing ac-cording to Beers criteria among elderly outpatients in Jordan: a cross sectional study." Pharmacy Practice (Granada) 17, no. 2 (2019).
  2. Al-Azzam, Sayer I., Belal A. Al-Husein, Firas Alzoubi, Majed M. Masadeh, and S. Al-Horani. "Self-medication with antibiotics in Jordanian population." International journal of occupational medicine and environmental health 20, no. 4 (2007): 373.

3.1. study area
Except for the location of the WWTP, the characteristics of the target point are unknown. Please add the characteristics of the target point.

Our response:

The location of biggest medical center in Jordan, WWTPs and groundwater spring is identified in Figure 1. In addition, the soils and plants samples are collected from the downstream farms that receive the reclaimed water (frames are not located in the map in order not cause any problem to farmers and government).

 In Table 3, what is the "MCM"?

Our response:

MCM means Million Cubic Meter and it is added to the table footer.

3.2. Sampling, sample preparation, and extraction
Please specify the sampling method, SPE method, and LC-MS/MS method. This is a very important part.

Our response:

Sampling, sample preparation, and extraction (for water, soil and plant samples) have been updated and referred to methods that provided by the Royal Scientific Society (RSS) in Jordan and Water Sciences Laboratory at the University of Nebraska–Lincoln (WSL/UNL) in the United States (USA) [Ref. No. 21]( Lines 159-160) and (lines 170-174).

Redraw Fig 2-4 in the same format.
Why is the graph divided into 3 parts and why are the substances divided?

Our response:

Figure 2 has been modified and formatted in one style. The parameters are different and combing them will make it too crowded.  Furthermore, the CBZ and DFC concentrations are important to show in separate figures.

4.3. PPCPs in cultivated farms
Add to "3.2", specifying "how to sample soil"

Our response:

Soil sampling and measurement  are mentioned in lines 172-175 (3.2 Sampling, Sample Preparation, and Extraction) . Soil pastes were prepared and filtered in order to prepare the soil extraction process was implemented according to the procedure provided by Water Sciences Laboratory at the University of Nebraska–Lincoln (WSL/UNL) in the United States (USA) [21].

Explain why CBZ concentrations are high in line 298.

Our response:

Line 128 : the CBZ consumption is high in Jordan, whereas, the CBZ removal rate is low by wastewater treatment plants explaining its high concentrations.

Why is only the CBZ concentration presented in Fig 6?

Our response:

CBZ showed high concentration in TWW, soils and plants.

Why was the surface layer of soil 20 cm and 40 cm presented?

Our response:

Soil sampling focuses on 20  and 40 cm depth that presents the area root of around 20 cm and trees can go down into 1 meter.

Why are all pharmaceuticals not presented in Fig 5?

Our response:

The concentration of other pharmaceuticals are very low as mentioned in results.  Only high PPCPs concentration are shown in Figure 5.

The "discussion" session should fully explain the parts of the "results" session. Please explain in detail the high and low concentration of pharmaceutical substances and the source of surrounding pollution.

Our response:

We thank the respected reviewer for the valuable input. The source of pollutants are the TWW effluents and no other sources are mentioned. We are afraid this will make the discussion redundant since all important results were also discussed  In the results part.

Why did Na'ur WWTP show high PPCPs removal efficiency in Table 5? explain it please.

Our response:

The Na`ur treatment process has an addition mechanical, chemical and biological  stages, whereas the other WWTPs has activated sludge and trickling filter treatment process (Table 3).

There are many typos throughout the paper. Please correct it.

Our response:

We thank the respected reviewer for this valuable input. All typos were corrected in the revised version of the manuscript.

Reviewer 2 Report

In this work the authors investigate the concentration of emerging waste water contaminants in Jordan. They discuss the source and fate, both of which are equally as important. The work is very well written. While the study takes places in Jordan, I believe the findings and discussion are important in general. The manuscript is of great value and is well written. I offer only a few minor comments.

Comments:

  1. Line 33: What are MENA countries? I would also suspect that this is broadly applicable.
  2. Line 34: What is TWW?
  3. Lines 71-72. This is an excellent point. It is often the perception that emerging contaminants are handled by the local waste water treatment plant. But this is not the case. The contaminants are often dilute, causing challenges. Second, the contaminants are often below therapeutic levels so that treatment in the past was not considered. In many countries long term exposure is becoming a concern, and this is beginning to be revisited.
  4. A general comment about the Introduction is it is very good. I have read similar studies which focus only on contaminant concentration. Here you focus on sources and implications which is of great value.
  5. Another general comment, could you provide the CAS number for the pharmaceuticals when appropriate? For some of the compounds it is likely that the common name used is different in other countries, and the CAS could help clarify.

Author Response

Authors thank you for your nice words,  comments and suggestions.

Line 33: What are MENA countries? I would also suspect that this is broadly applicable.

Our response:

Line 33: MENA countries mean Middle East and North Africa region. This shortcut has been clarified in the text and highlighted in red color.

Line 34: What is TWW?

Our response:

Line 34: TWW means Treated waste water. It has been mention before using its shortcut.

Lines 71-72. This is an excellent point. It is often the perception that emerging contaminants are handled by the local waste water treatment plant. But this is not the case. The contaminants are often dilute, causing challenges. Second, the contaminants are often below therapeutic levels so that treatment in the past was not considered. In many countries long term exposure is becoming a concern, and this is beginning to be revisited.

Our response:

We totally agree with your concern that and many thanks for supporting this point in the study.

A general comment about the Introduction is it is very good. I have read similar studies which focus only on contaminant concentration. Here you focus on sources and implications which is of great value.

Our response:

The Status que of the present pharmaceuticals and their impact is warning for future.  Your note is really appreciated.

Another general comment, could you provide the CAS number for the pharmaceuticals when appropriate? For some of the compounds it is likely that the common name used is different in other countries, and the CAS could help clarify.

Our response:

CAS number is not available to the authors.

Reviewer 3 Report

The paper is interesting.
I recommend publication only if the following issues can be addressed.

- The innovation of this research needs to be better explored. 

- Lines 31-42 page 1: You should mention that discharge of saline wastewater (e.g. pharmaceutical wastewater) degrades water quality and thus it cannot be directly used for potable water (via desalination) and industrial applications. Cite the following references:

Panagopoulos, A. (2020). Techno-economic evaluation of a solar multi‐effect distillation/thermal vapor compression hybrid system for brine treatment and salt recovery. Chemical Engineering and Processing - Process Intensification, 152. 

Panagopoulos, A. (2021). Techno-economic assessment of Minimal Liquid Discharge (MLD) treatment systems for saline wastewater (brine) management and treatment. Process Safety and Environmental Protection, 146, pp. 656-669. 

Panagopoulos, A. (2022). Techno-economic assessment of zero liquid discharge (ZLD) systems for sustainable treatment, minimization and valorization of seawater brine. Journal of Environmental Management, 306, 114488.

- Much more explanations and interpretations must be added for the Results

- How many replications you performed for your experiments?

- Conclusion: Include more of your results.

- Conclusion: Discuss the applicability of your findings/results and future study in this field.

- Language editing is recommended.

Author Response

Many thanks for your comments, notes and suggestions.

The innovation of this research needs to be better explored.

Our response:

The text, figures and tables have been modified in general.

Lines 31-42 page 1: You should mention that discharge of saline wastewater (e.g. pharmaceutical wastewater) degrades water quality and thus it cannot be directly used for potable water (via desalination) and industrial applications. Cite the following references:

Our response:

Page 1, line 31-42, the effluents from the mentioned WWTPs  in the study and from all the WWTPs in Jordan do not discharge  any kind of saline wastewater. The Maximum effluents electric conductivity is 2500 μs/cm during summer and conform to the Jordanian standards for reuse and discharge. No need for desalination process with reclaimed water in Jordan.  Therefore, all the suggested references are not suitable to cite.

Much more explanations and interpretations must be added for the Results

Our response:

Results have been modified and updated.

How many replications you performed for your experiments?

Our response:

The sampling has took two round containing 10 water, two groundwater,10 soil and 11 plant samples each time.  In the lab, two replications of each samples were analyzed.

Conclusion: Include more of your results. Conclusion: Discuss the applicability of your findings/results and future study in this field.

Our response:

Conclusion has been modified and updated. For future, line 349: Jordan by now need new laws, by-laws, and regulations to govern the production, distribution, use, collection, and proper disposal of unused pharmaceutical residues along the whole chain of producers, distributers, users and handling of the unused quantities.

Language editing is recommended.

Our response:

The language has been edited.

Round 2

Reviewer 1 Report

First, authors thank you for your comments, notes and suggestions.

Title and content do not match. What are "Other emerging pollutants"?

Our response:

The title of this study has been modified to “Pharmaceuticals as Emerging Pollutants in the Re-Claimed Wastewater used in Irrigation and their Effects on Plants, Soils, and Groundwater” in order match the study content. [OK]

Plants, soils, and Groundwater, Human, and Animal Health" doesn't seem to have any results.
I've seen the results of the soil, but other results are just speculation.

Our response:

We thank the respected reviewer for the point. Page 6, line 196, heading 4.2. PPCPs in WWTPs shows the concentrations of Pharmaceutical residues  in wastewater treatment plants, Boqorreya Spring (groundwater) and the mixed water [Salt outlet + Boqorreya Spring]. In addition, page 9,  line 261 shows 4.3. Results of PPCPs in Cultivated Farms, including soils and plants analyses (Figure 5, 6 and 7).[OK]

Abbreviations for pharmaceutical substances are only partially used. If you use abbreviations, apply them all.

Our response:

Abbreviations have been used in whole text in the revised version of the manuscript

If you want to use an abbreviation for pharmaceuticals, other pharmaceuticals must also use the abbreviation. otherwise, CBZ should be carbamazepine and DCF should be diclofenac.  

ex) ciprofloxacin --> ??, lincomycin --> LCM?? 

"Risk" was not assessed in this paper in line 80-88. Rewrite the "object" in this study.

Our response:

We thank the respected reviewer for the valuable suggestion.

Line 80 has been rephrased into the following:  The purpose of our study is to assess the existing PPCPs in Na`ur, Fuhais-Mahis, Abu- Nuseir, and Al Baqa`a wastewater treatment plants as major effluent discharge sites and tracking the mixing process along wadis with springs` water until reuse application in irrigation to assess the risk potential to the environment. Furthermore, soils, plants and surface and groundwater are part of the environment. So the presence of PPCPs has been assessed through this study.[OK]

In the title and purpose of the study, be sure to mention only the part that presents the results.

Our response:

We thank the respected reviewer for this valuable comment. The title of this study has been modified into “Pharmaceuticals as Emerging Pollutants in the Re-Claimed Wastewater used in Irrigation and their Effects on Plants, Soils, and Groundwater”. Thus, surface, groundwater, plants and soils analyses are mentioned in the title and results.[OK]

Add references to the sentences on lines 100-115.

Our response:

References (17 and 18) were added in the modified version of the manuscript.

  1. Al-Azayzih, Ahmad, Rawan Alamoori, and Shoroq M. Altawalbeh. "Potentially inappropriate medications prescribing ac-cording to Beers criteria among elderly outpatients in Jordan: a cross sectional study." Pharmacy Practice (Granada) 17, no. 2 (2019).
  2. Al-Azzam, Sayer I., Belal A. Al-Husein, Firas Alzoubi, Majed M. Masadeh, and S. Al-Horani. "Self-medication with antibiotics in Jordanian population." International journal of occupational medicine and environmental health 20, no. 4 (2007): 373.[OK]

3.1. study area
Except for the location of the WWTP, the characteristics of the target point are unknown. Please add the characteristics of the target point.

Our response:

The location of biggest medical center in Jordan, WWTPs and groundwater spring is identified in Figure 1. In addition, the soils and plants samples are collected from the downstream farms that receive the reclaimed water (frames are not located in the map in order not cause any problem to farmers and government).[OK]

 In Table 3, what is the "MCM"?

Our response:

MCM means Million Cubic Meter and it is added to the table footer.[OK]

3.2. Sampling, sample preparation, and extraction
Please specify the sampling method, SPE method, and LC-MS/MS method. This is a very important part.

Our response:

Sampling, sample preparation, and extraction (for water, soil and plant samples) have been updated and referred to methods that provided by the Royal Scientific Society (RSS) in Jordan and Water Sciences Laboratory at the University of Nebraska–Lincoln (WSL/UNL) in the United States (USA) [Ref. No. 21]( Lines 159-160) and (lines 170-174).[OK]

Redraw Fig 2-4 in the same format.
Why is the graph divided into 3 parts and why are the substances divided?

Our response:

Figure 2 has been modified and formatted in one style. The parameters are different and combing them will make it too crowded.  Furthermore, the CBZ and DFC concentrations are important to show in separate figures.[OK]

4.3. PPCPs in cultivated farms
Add to "3.2", specifying "how to sample soil"

Our response:

Soil sampling and measurement  are mentioned in lines 172-175 (3.2 Sampling, Sample Preparation, and Extraction) . Soil pastes were prepared and filtered in order to prepare the soil extraction process was implemented according to the procedure provided by Water Sciences Laboratory at the University of Nebraska–Lincoln (WSL/UNL) in the United States (USA) [21].[OK]

Explain why CBZ concentrations are high in line 298.

Our response:

Line 128 : the CBZ consumption is high in Jordan, whereas, the CBZ removal rate is low by wastewater treatment plants explaining its high concentrations.

Are there any other CBZ sources besides WWTP? Is non-point pollution taken into account?

Why is only the CBZ concentration presented in Fig 6?

Our response:

CBZ showed high concentration in TWW, soils and plants.[OK]

Why was the surface layer of soil 20 cm and 40 cm presented?

Our response:

Soil sampling focuses on 20  and 40 cm depth that presents the area root of around 20 cm and trees can go down into 1 meter.[OK]

Why are all pharmaceuticals not presented in Fig 5?

Our response:

The concentration of other pharmaceuticals are very low as mentioned in results.  Only high PPCPs concentration are shown in Figure 5.[OK]

The "discussion" session should fully explain the parts of the "results" session. Please explain in detail the high and low concentration of pharmaceutical substances and the source of surrounding pollution.

Our response:

We thank the respected reviewer for the valuable input. The source of pollutants are the TWW effluents and no other sources are mentioned. We are afraid this will make the discussion redundant since all important results were also discussed  In the results part.[OK]

Why did Na'ur WWTP show high PPCPs removal efficiency in Table 5? explain it please.

Our response:

The Na`ur treatment process has an addition mechanical, chemical and biological  stages, whereas the other WWTPs has activated sludge and trickling filter treatment process (Table 3).[OK]

There are many typos throughout the paper. Please correct it.

Our response:

We thank the respected reviewer for this valuable input. All typos were corrected in the revised version of the manuscript.[OK]

Author Response

Dear respected Editor of Water,

Thank you for your letter and constructive comments concerning our manuscript entitled Pharmaceuticals as Emerging Pollutants in the Re-Claimed Wastewater used in Irrigation and their Effects on Plants, Soils, and Groundwater .

We have studied your second comments made corrections accordingly and we hope would meet with your approval. Please consider our responses in the following texts.

Response to Reviewer 1:

If you want to use an abbreviation for pharmaceuticals, other pharmaceuticals must also use the abbreviation. otherwise, CBZ should be carbamazepine and DCF should be diclofenac.  

  1. ex) ciprofloxacin --> ??, lincomycin --> LCM?? 

Our response:

Abbreviations of pharmaceuticals have been replaced via their names without using the abbreviation in the whole text.

Are there any other CBZ sources besides WWTP? Is non-point pollution taken into account?

Our response:

Effluents from various WWTPs are the only source of CBZ.

Many Thanks

Author Response

Dear respected Editor of Water,

Thank you for your letter and constructive comments concerning our manuscript entitled Pharmaceuticals as Emerging Pollutants in the Re-Claimed Wastewater used in Irrigation and their Effects on Plants, Soils, and Groundwater .

We have studied your second comments made corrections accordingly and we hope would meet with your approval. 

The English has been checked and improved. 

Many thanks!

Authors

Round 3

Reviewer 1 Report

I will accept it in this form. But I would strongly recommend that the authors re-examine the "Journal Format".